# New wave behaviors of the Fokas-Lenells model using three integration techniques

**Mohammad Safi Ullah**[1,3]*, **Harun-Or Roshid**[2], **M. Zulfikar Ali**[3]

**1** Department of Mathematics, Comilla University, Cumilla, Bangladesh, **2** Department of Mathematics, Pabna University of Science and Technology, Pabna, Bangladesh, **3** Department of Mathematics, University of Rajshahi, Rajshahi, Bangladesh

* safi.ru1985@gmail.com

**Data Availability Statement:** All relevant data can be found in the paper.

**Funding:** The author(s) received no specific funding for this work.

## Abstract

In this investigation, we apply the improved Kudryashov, the novel Kudryashov, and the unified methods to demonstrate new wave behaviors of the Fokas-Lenells nonlinear waveform arising in birefringent fibers. Through the application of these techniques, we obtain numerous previously unreported novel dynamic optical soliton solutions in mixed hyperbolic, trigonometric, and rational forms of the governing model. These solutions encompass periodic waves with W-shaped profiles, gradually increasing amplitudes, rapidly increasing amplitudes, double-periodic waves, and breather waves with symmetrical or asymmetrical amplitudes. Singular solitons with single and multiple breather waves are also derived. Based on these findings, we can say that our implemented methods are more reliable and useful when retrieving optical soliton results for complicated nonlinear systems. Various potential features of the derived solutions are presented graphically.

## 1 Introduction

In the telecommunications industry, solitons are one of the fastest-growing study fields. Without the idea of a solitary wave, it is not easy to understand how fiber optics [1, 2], telephones [3, 4], and many other types of communication [5–7] operate. Consequently, mathematical physicists are highly interested in obtaining soliton solutions for nonlinear models [8, 9]. Numerous well-known nonlinear structures exhibit the presence of soliton solutions, such as the Jimbo-Miwa-like model [10], the STOL model [11], the Sine-Gordon equation [12], the nonlinear Schrödinger equation [13], the Konopelchenko-Dubrovsky equation [14, 15], the nonlocal Klein-Gordon model [16], the Wadati-Konno-Ichikawa equation [17], the phi-four model [18], the Gerdjikov-Ivanov equation [19], the deoxyribonucleic acid (DNA) model [20], the KdV-BBM equation [21], etc. Diverse effective methods exist for managing these nonlinear structures and deriving soliton solutions, such as the generalized exponential rational function technique [22, 23], the generalized Riccati equation mapping scheme [24], the ð-dressing method [25], the extended sinh-Gordon equation approach [26], the Hirota bilinear technique [27], etc.

A renowned model, known as the Fokas-Lenells model, was first presented in 2009 [28] and has since achieved various honors. There have been numerous applications for this model, including fiber optics. The mentioned system can be used to describe the dynamic features of

**Competing interests:** The authors have declared that no competing interests exist.

optical and photonic crystal fibers [29]. To express soliton solutions, the Fokas-Lenells PDE has been studied by applying numerous reliable and effective elegant algorithms, such as the Ricati equation scheme [30], extended trial equation scheme [31], complex envelope function ansatz [29], mapping scheme [32], unified solver method [33], $\varphi^6$-model expansion approach [34], etc.

The prime focus of this research is to present the novel wave behaviors of the suggested model using three integration schemes, the unified method [35, 36], the improved Kudryashov method [37], and the novel Kudryashov technique [38, 39] which can describe the dynamic feature of the Fokas-Lenells model.

This research literature is formulated as underneath: Section 2 contains the operating model. An ordinary differential form of the operating model can be found in Section 3. Sections 4, 5, and 6 summarize and implement the unified method, the improved Kudryashov method, and the novel Kudryashov technique, respectively. Section 7 describes the graphic analysis with a discussion of the operating model's solutions. Finally, a summary of the article and plans for future research are given in Section 8.

We confirm that the optical soliton results of the Fokas-Lenells dynamical waveform obtained through our employed methods are the first reported and have not been studied until now.

## 2 Operating model

The dimensionless Fokas-Lenells PDE has the following form [28–34]:

$$iQ_t + a_1 Q_{xx} + a_2 Q_{xt} + (bQ + isQ_x)|Q|^2 = i[aQ_x + l(|Q|^{2n}Q)_x \\ + m(|Q|^{2n})_x Q]. \tag{2.1}$$

In the aforementioned model, $Q(x, t)$ denotes the wave's magnitude with distance coordinate $x$ and time coordinate $t$, where $i = \sqrt{-1}$. The potentials $a$, $a_1$, $a_2$, and $s$ signify the coefficient of inter-model dispersion, GVD, STD, and nonlinear dispersion, sequentially. The potentials $b$, $n$, and $l$ signify self-phase modulation, the effect of full non-linearity, and the self-steepening effect, sequentially. Finally, $m$ is another type of nonlinear dispersion. It's important to note that the parameter $m$ holds a real numerical value. Nonetheless, if $m$ is entirely imaginary, it would depict Raman scattering. This phenomenon contributes to the frequency shift of solitons and is characterized by the dissipative Raman effect. We can say that, the Fokas-Lenells model is a comprehensive equation that combines dispersion, nonlinearity, and various effects to describe the evolution of a wave's magnitude $Q(x, t)$. The coefficients $a$, $a_1$, $a_2$, $s$, $m$, $b$, $n$, and $l$ play distinct roles in capturing the influence of different physical phenomena on the wave's behavior.

## 3 ODE formulation of the model

To resolve equation Eq (2.1), we will take into account the following solution structure:

$$Q(x, t) = U(\varsigma) \exp(i\delta), \tag{3.1}$$

whilst $\varsigma = x - gt$ with velocity component $g$ in which the phase component $\delta = -kx + wt + p$ and the amplitude component $U$ whereas wave number $w$, frequency $k$, and phase value $p$. Now by utilizing Eqs (2.1) and (3.1), the imaginary and real portions will be separated. Then the real part is

$$(a_1 - a_2 g)U'' + (a_2 wk - a_1 k^2 - w - ak)U + (ks + b)U^3 - kU[2mn \\ + (2n+1)l]U^{2n} = 0. \tag{3.2}$$

and, the imaginary portion implies

$$g + 2ka_1 - a_2(gk + w) + a - sU^2 + [2mn + l(2n + 1)]U^{2n} = 0. \tag{3.3}$$

From Eq (3.3), we have

$$[2nm + (2n + 1)l]U^{2n} = -g - 2ka_1 + a_2(gk + w) - a + sU^2. \tag{3.4}$$

In Eq (3.4) by plugging $[2mn + l(2n + 1)]U^{2n} = 0$, $s = 0$, one reaches

$$g = \frac{2ka_1 - a_2w + a}{a_2k - 1}, l = \frac{-2mn}{2n + 1}. \tag{3.5}$$

Accordingly, from Eq (3.2) we have

$$(a_1 - a_2g)U'' + (a_1k^2 - w + kg(1 - a_2k))U + bU^3 = 0. \tag{3.6}$$

## 4 Description of the unified technique with application

Let the auxiliary solution of Eq (3.6) be [35, 36]

$$U(\varsigma) = l_0 + \sum_{i=1}^{N}[l_i\Psi(\varsigma)^i + m_i\Psi(\varsigma)^{-i}], \tag{4.1}$$

$$\Psi'(\varsigma) = \Psi^2(\varsigma) + \vartheta. \tag{4.2}$$

There are nine possible solutions to equation Eq (4.2) in three different families:
**Family-01:** Hyperbolic function (for $\vartheta < 0$):

$$\Psi(\varsigma) = \begin{cases} \frac{\sqrt{-(G^2+H^2)\vartheta}-G\sqrt{-\vartheta}\cosh(2\sqrt{-\vartheta}(\varsigma+\rho))}{G\sinh(2\sqrt{-\vartheta}(\varsigma+\rho))+H}, \\ \frac{-\sqrt{-(G^2+H^2)l}-G\sqrt{-\vartheta}\cosh(2\sqrt{-\vartheta}(\varsigma+\rho))}{G\sinh(2(\varsigma+\rho)\sqrt{-\vartheta})+H}, \\ \sqrt{-\vartheta} + \frac{2G\sqrt{-\vartheta}}{G+\cosh(2\sqrt{-\vartheta}(\varsigma+\rho))-\sinh(2\sqrt{-\vartheta}(\varsigma+\rho))}, \\ -\sqrt{-\vartheta} + \frac{2G\sqrt{-\vartheta}}{G+\cosh(2\sqrt{-\vartheta})(\varsigma+\rho)-\sinh(2\sqrt{-\vartheta}(\varsigma+\rho))}, \end{cases} \tag{4.3}$$

**Family-02:** Trigonometric function (for $\vartheta > 0$):

$$\Psi(\varsigma) = \begin{cases} \frac{\sqrt{(G^2-H^2)\vartheta}-G\sqrt{\vartheta}\cos(2\sqrt{\vartheta}(\varsigma+\rho))}{G\sin(2\sqrt{\vartheta}(\varsigma+\rho))+H}, \\ \frac{-\sqrt{(G^2-H^2)\vartheta}-G\sqrt{\vartheta}\cos(2\sqrt{\vartheta}(\varsigma+\rho))}{G\sin(2\sqrt{\vartheta}(\varsigma+\rho))+H}, \\ i\sqrt{\vartheta} + \frac{-2iG\sqrt{\vartheta}}{G+\cos(2\sqrt{\vartheta}(\varsigma+\rho))-i\sin(2\sqrt{\vartheta}(\varsigma+\rho))}, \\ -i\sqrt{\vartheta} + \frac{2iG\sqrt{\vartheta}}{G+\cos(2\sqrt{\vartheta}(\varsigma+\rho))-i\sin(2\sqrt{\vartheta}(\varsigma+\rho))}, \end{cases} \tag{4.4}$$

**Family-03:** Rational function (for $\vartheta$ equal zero)

$$\Psi(\varsigma) = -\frac{1}{\varsigma + \rho}, \tag{4.5}$$

where $\rho$, $H$, and $G \neq 0$ are real parameters. For finding $N$ in Eq (4.1), balance between $U^3$ and

$U''$ yields $N = 1$. Then Eq (4.1) will be converted as

$$U(\varsigma) = l_0 + l_1 \Psi(\varsigma) + m_1 \Psi(\varsigma)^{-1}. \tag{4.6}$$

Now, making use of Eqs (4.6), (4.2) and (3.6) and some simple calculation gives

$$\begin{cases} l_0 = 0, l_1 = l_1, m_1 = 0, a = -\frac{bk^2 a_2^2 l_1^2 + 2b\vartheta a_2^2 l_1^2 - 2bka_2 l_1^2 + bl_1^2 + 2a_1}{2a_2}, \\ b = b, k = k, w = -\frac{bk^2 a_2 l_1^2 + 2b\vartheta a_2 l_1^2 - bkl_1^2 - 2ka_1}{2a_2}, g = \frac{2ka_1 - a_2 w + a}{a_2 k - 1}, \end{cases} \tag{4.7}$$

$$\begin{cases} l_0 = 0, l_1 = 0, m_1 = m_1, a = \frac{bk^2 a_2^2 m_1^2 + 2b\vartheta a_2^2 m_1^2 - 2bka_2 m_1^2 + bm_1^2 - 2\vartheta^2 a_1}{2\vartheta^2 a_2}, \\ b = b, k = k, w = \frac{bk^2 a_2 m_1^2 + 2b\vartheta a_2 m_1^2 - bkm_1^2 + 2k\vartheta^2 a_1}{2\vartheta^2 a_2}, g = \frac{2ka_1 - a_2 w + a}{a_2 k - 1}, \end{cases} \tag{4.8}$$

$$\begin{cases} l_0 = 0, l_1 = \frac{\sqrt{m_1}}{\vartheta}, m_1 = m_1, a = \frac{bk^2 a_2^2 m_1^2 + \sqrt{-16b\vartheta a_2^2 m_1^2} + 2b\vartheta a_2^2 m_1^2 - 2bka_2 m_1^2 + bm_1^2 - 2\vartheta^2 a_1}{2\vartheta^2 a_2}, \\ b = b, k = k, w = \frac{bk^2 a_2 m_1^2 + \sqrt{-16b\vartheta a_2 m_1^2} + 2b\vartheta a_2 m_1^2 - bkm_1^2 + 2k\vartheta^2 a_1}{2\vartheta^2 a_2}, g = \frac{2ka_1 - a_2 w + a}{a_2 k - 1}. \end{cases} \tag{4.9}$$

Applying Eq (3.1) and Eqs (4.3)–(4.5), by the aid of solution Eq (4.7) gives the next 9 exact solutions of Eq (2.1).

$$Q_1(x, t) = l_1 \left( \frac{\sqrt{-(G^2 + H^2)\vartheta} - G\sqrt{-\vartheta} \cosh\left(2\sqrt{-\vartheta}(\varsigma + \rho)\right)}{G \sinh\left(2\sqrt{-\vartheta}(\varsigma + \rho)\right) + H} \right) \times \exp(i\delta),$$

$$Q_2(x, t) = l_1 \left( \frac{-\sqrt{-(G^2 + H^2)\vartheta} - G\sqrt{-\vartheta} \cosh\left(2\sqrt{-\vartheta}(\varsigma + \rho)\right)}{G \sinh\left(2\sqrt{-\vartheta}(\varsigma + \rho)\right) + H} \right) \times \exp(i\delta),$$

$$Q_3(x, t) = l_1 \left( \sqrt{-\vartheta} + \frac{2G\sqrt{-\vartheta}}{G + \cosh\left(2(\varsigma + \rho)\sqrt{-\vartheta}\right) - \sinh\left(2(\varsigma + \rho)\sqrt{-\vartheta}\right)} \right) \times \exp(i\delta),$$

$$Q_4(x, t) = l_1 \left( -\sqrt{-\vartheta} + \frac{2G\sqrt{-\vartheta}}{G + \cosh\left(2(\varsigma + \rho)\sqrt{-\vartheta}\right) - \sinh\left(2(\varsigma + \rho)\sqrt{-\vartheta}\right)} \right) \times \exp(i\delta),$$

$$Q_5(x, t) = l_1 \left( \frac{\sqrt{(G^2 - H^2)\vartheta} - G\sqrt{\vartheta}\cos(2\sqrt{\vartheta}(\varsigma + \rho))}{G\sin(2\sqrt{\vartheta}(\varsigma + \rho)) + H} \right) \times \exp(i\delta),$$

$$Q_6(x, t) = l_1 \left( \frac{-\sqrt{(G^2 - H^2)\vartheta} - G\sqrt{\vartheta}\cos(2\sqrt{\vartheta}(\varsigma + \rho))}{G\sin(2\sqrt{\vartheta}(\varsigma + \rho)) + H} \right) \times \exp(i\delta),$$

$$Q_7(x, t) = l_1 \left( i\sqrt{\vartheta} + \frac{-2iG\sqrt{\vartheta}}{G + \cos(2\sqrt{\vartheta}(\varsigma + \rho)) - i\sin(2\sqrt{\vartheta}(\varsigma + \rho))} \right) \times \exp(i\delta),$$

$$Q_8(x, t) = l_1 \left( -i\sqrt{\vartheta} + \frac{2iG\sqrt{\vartheta}}{G + \cos(2\sqrt{\vartheta}(\varsigma + \rho)) - i\sin(2\sqrt{\vartheta}(\varsigma + \rho))} \right) \times \exp(i\delta),$$

$$Q_9(x, t) = -\frac{l_1}{\varsigma + \rho} \times \exp(i\delta),$$

where $\delta = -kx + wt + p$. Applying Eq (3.1) and Eqs (4.3)–(4.5), by the aid of solution Eq (4.8)

gives the next 8 exact solutions of Eq (2.1).

$$Q_{10}(x, t) = m_1 \left( \frac{\sqrt{-(G^2 + H^2)\vartheta} - G\sqrt{-\vartheta} \cosh\left(2\sqrt{-\vartheta}(\varsigma + \rho)\right)}{G \sinh\left(2\sqrt{-\vartheta}(\varsigma + \rho)\right) + H} \right)^{-1} \times \exp(i\delta),$$

$$Q_{11}(x, t) = m_1 \left( \frac{-\sqrt{-(G^2 + H^2)\vartheta} - G\sqrt{-\vartheta} \cosh\left(2\sqrt{-\vartheta}(\varsigma + \rho)\right)}{G \sinh\left(2\sqrt{-\vartheta}(\varsigma + \rho)\right) + H} \right)^{-1} \times \exp(i\delta),$$

$$Q_{12}(x, t) = m_1 \left( \sqrt{-\vartheta} + \frac{2G\sqrt{-\vartheta}}{G + \cosh\left(2(\varsigma + \rho)\sqrt{-\vartheta}\right) - \sinh\left(2(\varsigma + \rho)\sqrt{-\vartheta}\right)} \right)^{-1} \times \exp(i\delta),$$

$$Q_{13}(x, t) = m_1 \left( -\sqrt{-\vartheta} + \frac{2G\sqrt{-\vartheta}}{G + \cosh\left(2(\varsigma + \rho)\sqrt{-\vartheta}\right) - \sinh\left(2(\varsigma + \rho)\sqrt{-\vartheta}\right)} \right)^{-1} \times \exp(i\delta),$$

$$Q_{14}(x, t) = m_1 \left( \frac{\sqrt{(G^2 - H^2)\vartheta} - G\sqrt{\vartheta}\cos(2\sqrt{\vartheta}(\varsigma + \rho))}{G\sin(2\sqrt{\vartheta}(\varsigma + \rho)) + H} \right)^{-1} \times \exp(i\delta),$$

$$Q_{15}(x, t) = m_1 \left( \frac{-\sqrt{(G^2 - H^2)\vartheta} - G\sqrt{\vartheta}\cos(2\sqrt{\vartheta}(\varsigma + \rho))}{G\sin(2\sqrt{\vartheta}(\varsigma + \rho)) + H} \right)^{-1} \times \exp(i\delta),$$

$$Q_{16}(x, t) = m_1 \left( i\sqrt{\vartheta} + \frac{-2iG\sqrt{\vartheta}}{G + \cos(2\sqrt{\vartheta}(\varsigma + \rho)) - i\sin(2\sqrt{\vartheta}(\varsigma + \rho))} \right)^{-1} \times \exp(i\delta),$$

$$Q_{17}(x, t) = m_1 \left( -i\sqrt{\vartheta} + \frac{2iG\sqrt{\vartheta}}{G + \cos(2\sqrt{\vartheta}(\varsigma + \rho)) - i\sin(2\sqrt{\vartheta}(\varsigma + \rho))} \right)^{-1} \times \exp(i\delta),$$

where $\delta = -kx + wt + p$. Applying Eq (3.1) and Eqs (4.3)–(4.5), by the aid of solution Eq (4.9) gives the next 8 exact solutions of Eq (2.1).

$$Q_{18}(x, t) = \left( l_1 \left( \frac{\sqrt{-(G^2 + H^2)\vartheta} - G\sqrt{-\vartheta} \cosh\left(2\sqrt{-\vartheta}(\varsigma + \rho)\right)}{G \sinh\left(2\sqrt{-\vartheta}(\varsigma + \rho)\right) + H} \right) \right.$$
$$\left. + m_1 \left( \frac{\sqrt{-(G^2 + H^2)\vartheta} - G\sqrt{-\vartheta} \cosh\left(2\sqrt{-\vartheta}(\varsigma + \rho)\right)}{G \sinh\left(2\sqrt{-\vartheta}(\varsigma + \rho)\right) + H} \right)^{-1} \right) \times \exp(i\delta),$$

$$Q_{19}(x, t) = \left( l_1 \left( \frac{-\sqrt{-(G^2 + H^2)\vartheta} - G\sqrt{-\vartheta} \cosh\left(2\sqrt{-\vartheta}(\varsigma + \rho)\right)}{G \sinh\left(2\sqrt{-\vartheta}(\varsigma + \rho)\right) + H} \right) \right.$$
$$\left. + m_1 \left( \frac{-\sqrt{-(G^2 + H^2)\vartheta} - G\sqrt{-\vartheta} \cosh\left(2\sqrt{-\vartheta}(\varsigma + \rho)\right)}{G \sinh\left(2\sqrt{-\vartheta}(\varsigma + \rho)\right) + H} \right)^{-1} \right) \times \exp(i\delta),$$

$$Q_{20}(x, t) = \left( l_1 \left( \sqrt{-\vartheta} + \frac{2G\sqrt{-\vartheta}}{G + \cosh\left(2(\varsigma + \rho)\sqrt{-\vartheta}\right) - \sinh\left(2(\varsigma + \rho)\sqrt{-\vartheta}\right)} \right) \right.$$
$$\left. + m_1 \left( \sqrt{-\vartheta} + \frac{2G\sqrt{-\vartheta}}{G + \cosh\left(2(\varsigma + \rho)\sqrt{-\vartheta}\right) - \sinh\left(2(\varsigma + \rho)\sqrt{-\vartheta}\right)} \right)^{-1} \right) \times \exp(i\delta),$$

$$Q_{21}(x, t) = \left( l_1 \left( -\sqrt{-\vartheta} + \frac{2G\sqrt{-\vartheta}}{G + \cosh\left(2(\varsigma + \rho)\sqrt{-\vartheta}\right) - \sinh\left(2(\varsigma + \rho)\sqrt{-\vartheta}\right)} \right) \right.$$
$$\left. + m_1 \left( -\sqrt{-\vartheta} + \frac{2G\sqrt{-\vartheta}}{G + \cosh\left(2(\varsigma + \rho)\sqrt{-\vartheta}\right) - \sinh\left(2(\varsigma + \rho)\sqrt{-\vartheta}\right)} \right)^{-1} \right) \times \exp(i\delta),$$

$$Q_{22}(x,t) = (l_1(\frac{\sqrt{(G^2-H^2)\vartheta}-G\sqrt{\vartheta}\cos(2\sqrt{\vartheta}(\varsigma+\rho))}{G\sin(2\sqrt{\vartheta}(\varsigma+\rho))+H})$$
$$+ m_1(\frac{\sqrt{(G^2-H^2)\vartheta}-G\sqrt{\vartheta}\cos(2\sqrt{\vartheta}(\varsigma+\rho))}{G\sin(2\sqrt{\vartheta}(\varsigma+\rho))+H})^{-1}) \times \exp(i\delta),$$

$$Q_{23}(x,t) = (l_1(\frac{-\sqrt{(G^2-H^2)\vartheta}-G\sqrt{\vartheta}\cos(2\sqrt{\vartheta}(\varsigma+\rho))}{G\sin(2\sqrt{\vartheta}(\varsigma+\rho))+H})$$
$$+ m_1(\frac{-\sqrt{(G^2-H^2)\vartheta}-G\sqrt{\vartheta}\cos(2\sqrt{\vartheta}(\varsigma+\rho))}{G\sin(2\sqrt{\vartheta}(\varsigma+\rho))+H})^{-1}) \times \exp(i\delta),$$

$$Q_{24}(x,t) = (l_1(i\sqrt{\vartheta}+\frac{-2iG\sqrt{\vartheta}}{G+\cos(2\sqrt{\vartheta}(\varsigma+\rho))-i\sin(2\sqrt{\vartheta}(\varsigma+\rho))})$$
$$+ m_1(i\sqrt{\vartheta}+\frac{-2iG\sqrt{\vartheta}}{G+\cos(2\sqrt{\vartheta}(\varsigma+\rho))-i\sin(2\sqrt{\vartheta}(\varsigma+\rho))})^{-1}) \times \exp(i\delta),$$

$$Q_{25}(x,t) = (l_1(-i\sqrt{\vartheta}+\frac{2iG\sqrt{\vartheta}}{G+\cos(2\sqrt{\vartheta}(\varsigma+\rho))-i\sin(2\sqrt{\vartheta}(\varsigma+\rho))})$$
$$+ m_1(-i\sqrt{\vartheta}+\frac{2iG\sqrt{\vartheta}}{G+\cos(2\sqrt{\vartheta}(\varsigma+\rho))-i\sin(2\sqrt{\vartheta}(\varsigma+\rho))})^{-1}) \times \exp(i\delta),$$

where $\delta = -kx + wt + p$.

## 5 Description of the improved Kudryashov method with application

Let the auxiliary solution of the suggested nonlinear structure as follows [37]

$$U(\varsigma) = \frac{\sum_{i=0}^{N} l_i \Psi(\varsigma)^i}{\sum_{j=0}^{K} q_i \Psi(\varsigma)^i}, \qquad (5.1)$$

$$\Psi'(\varsigma) = M - \Psi^2(\varsigma). \qquad (5.2)$$

There are five possible solutions to equation Eq (5.2) in three different families:
**Family-01:** Hyperbolic function (for $M > 0$):

$$\Psi(\varsigma) = \begin{cases} \sqrt{M} \tanh(\sqrt{M}\varsigma), \\ \sqrt{M} \coth(\sqrt{M}\varsigma), \end{cases} \qquad (5.3)$$

**Family-02:** Trigonometric function (for $M < 0$):

$$\Psi(\varsigma) = \begin{cases} -\sqrt{-M} \tanh(\sqrt{-M}\varsigma), \\ \sqrt{-M} \coth(\sqrt{-M}\varsigma), \end{cases} \qquad (5.4)$$

**Family-03:** Rational function (for $M = 0$)

$$\Psi(\varsigma) = \frac{1}{\varsigma}. \qquad (5.5)$$

For finding $N$ in Eq (5.1), by balancing $U^3$ and $U''$ yields $N = K + 1$. If $K = 1$, then $N = 2$ and Eq (5.1) can be given in the formation

$$U(\varsigma) = \frac{l_0 + l_1 \Psi(\varsigma) + l_2 \Psi(\varsigma)^2}{q_0 + q_1 \Psi(\varsigma)}. \tag{5.6}$$

Now using Eqs (5.6), (5.2), and (3.6) and some simple calculation gives

$$
\begin{cases}
w = \dfrac{a_1(-k^3 a_2 + 2Mka_2 + k^2 + 2M)}{(-k^2 a_2^2 + 2Ma_2^2 + 2ka_2 - 1)}, l_0 = \dfrac{\sqrt{2b(-k^2 a_2^2 + 2Ma_2^2 + 2ka_2 - 1)a_1}}{(b(-k^2 a_2^2 + 2Ma_2^2 + 2ka_2 - 1))} q_1 M, a = a, \\[4mm]
l_1 = \dfrac{\sqrt{2b(-k^2 a_2^2 + 2Ma_2^2 + 2ka_2 - 1)a_1}}{b(-k^2 a_2^2 + 2Ma_2^2 + 2ka_2 - 1)} q_0, l_2 = 0, q_0 = q_0, q_1 = q_1, g = \dfrac{2ka_1 - a_2 w + a}{a_2 k - 1},
\end{cases}
\tag{5.7}
$$

$$
\begin{cases}
w = \dfrac{a_1(-k^3 a_2 + 8Mka_2 + k^2 + 8M)}{(-k^2 a_2^2 + 8Ma_2^2 + 2ka_2 - 1)}, l_0 = l_2 M, l_1 = 0, l_2 = l_2, a = a, \\[4mm]
q_1 = \dfrac{\sqrt{(-k^2 a_2^2 + 8Ma_2^2 + 2ka_2 - 1)ba_1}}{\sqrt{2}a_1} l_2, q_0 = q_0, g = \dfrac{2ka_1 - a_2 w + a}{a_2 k - 1},
\end{cases}
\tag{5.8}
$$

$$
\begin{cases}
w = \dfrac{a_1(-k^3 a_2 + 2Mka_2 + k^2 + 2M)}{-k^2 a_2^2 + 2Ma_2^2 + 2ka_2 - 1}, l_0 = 0, l_1 = \dfrac{a_1 q_0 \sqrt{2}}{\sqrt{ba_1(-k^2 a_2^2 + 2Ma_2^2 + 2ka_2 - 1)}}, a = a, \\[4mm]
l_2 = \dfrac{\sqrt{2ba_1(-k^2 a_2^2 + 2Mka_2^2 + 2ka_2 - 1)}}{b(-k^2 a_2^2 + 2Ma_2^2 + 2ka_2 - 1)} q_1, q_0 = q_0, q_1 = q_1, g = \dfrac{2ka_1 - a_2 w + a}{a_2 k - 1}.
\end{cases}
\tag{5.9}
$$

Applying Eqs (5.3)–(5.5) and Eq (3.1), by the aid of the solution Eq (5.7) gives the next 5 exact solutions Eq (2.1).

$$Q_{26}(x, t) = \frac{l_0 + l_1 \sqrt{M} \tanh(\sqrt{M}\varsigma)}{q_0 + q_1 \sqrt{M} \tanh(\sqrt{M}\varsigma)} \times \exp(i\delta),$$

$$Q_{27}(x, t) = \frac{l_0 + l_1 \sqrt{M} \coth(\sqrt{M}\varsigma)}{q_0 + q_1 \sqrt{M} \coth(\sqrt{M}\varsigma)} \times \exp(i\delta),$$

$$Q_{28}(x, t) = \frac{l_0 - l_1 \sqrt{-M} \tan(\sqrt{-M}\varsigma)}{q_0 - q_1 \sqrt{-M} \tan(\sqrt{-M}\varsigma)} \times \exp(i\delta),$$

$$Q_{29}(x, t) = \frac{l_0 + l_1 \sqrt{-M} \cot(\sqrt{-M}\varsigma)}{q_0 + q_1 \sqrt{-M} \cot(\sqrt{-M}\varsigma)} \times \exp(i\delta),$$

$$Q_{30}(x, t) = \frac{l_0 \varsigma + l_1}{q_0 \varsigma + q_1} \times \exp(i\delta),$$

where $\delta = -kx + wt + p$. Applying Eqs (5.3)–(5.5) and Eq (3.1), by the aid of the solution Eq

(5.8) gives the next 5 exact outcomes of Eq (2.1).

$$Q_{31}(x, t) = \frac{l_0 + l_2(\sqrt{M} \tanh(\sqrt{M}\varsigma))^2}{q_1\sqrt{M} \tanh(\sqrt{M}\varsigma)} \times \exp(i\delta),$$

$$Q_{32}(x, t) = \frac{l_0 + l_2(\sqrt{M} \coth(\sqrt{M}\varsigma))^2}{q_1\sqrt{M} \coth(\sqrt{M}\varsigma)} \times \exp(i\delta),$$

$$Q_{33}(x, t) = \frac{l_0 + l_2(\sqrt{-M}\tan(\sqrt{-M}\varsigma))^2}{-q_1\sqrt{-M}\tan(\sqrt{-M}\varsigma)} \times \exp(i\delta),$$

$$Q_{34}(x, t) = \frac{l_0 + l_2(\sqrt{-M}\cot(\sqrt{-M}\varsigma))^2}{q_1\sqrt{-M}\cot(\sqrt{-M}\varsigma)} \times \exp(i\delta),$$

$$Q_{35}(x, t) = \frac{l_0\varsigma^2 + l_2}{q_1\varsigma} \times \exp(i\delta),$$

where $\delta = -kx + wt + p$. Applying Eqs (5.3)–(5.5) and Eq (3.1), by the aid of the solution Eq (5.9) gives the next 5 exact solutions of Eq (2.1).

$$Q_{36}(x, t) = \frac{l_1\sqrt{M} \tanh(\sqrt{M}\varsigma) + l_2(\sqrt{M} \tanh(\sqrt{M}\varsigma))^2}{q_0 + q_1\sqrt{M} \tanh(\sqrt{M}\varsigma)} \times \exp(i\delta),$$

$$Q_{37}(x, t) = \frac{l_1\sqrt{M} \coth(\sqrt{M}\varsigma) + l_2(\sqrt{M} \coth(\sqrt{M}\varsigma))^2}{q_0 + q_1\sqrt{M} \coth(\sqrt{M}\varsigma)} \times \exp(i\delta),$$

$$Q_{38}(x, t) = \frac{-l_1\sqrt{-M}\tan(\sqrt{-M}\varsigma) + l_2(\sqrt{-M}\tan(\sqrt{-M}\varsigma))^2}{q_0 - q_1\sqrt{-M}\tan(\sqrt{-M}\varsigma)} \times \exp(i\delta),$$

$$Q_{39}(x, t) = \frac{l_1\sqrt{-M}\cot(\sqrt{-M}\varsigma) + l_2(\sqrt{-M}\cot(\sqrt{-M}\varsigma))^2}{q_0 + q_1\sqrt{-M}\cot(\sqrt{-M}\varsigma)} \times \exp(i\delta),$$

$$Q_{40}(x, t) = \frac{l_1\varsigma + l_2}{q_0\varsigma^2 + q_1\varsigma} \times \exp(i\delta),$$

where $\delta = -kx + wt + p$.

## 6 Description of the novel Kudryashov method with application

The auxiliary solution to the suggested nonlinear structure is assumed in the underneath symbolic form [38, 39]

$$U(\varsigma) = \sum_{i=0}^{K} A_i \Psi(\varsigma)^i, \tag{6.1}$$

$$\Psi'(\varsigma) = \sqrt{\Psi(\varsigma)^2(1 - 4LM\Psi(\varsigma)^2)}, \tag{6.2}$$

From equation Eq (6.2), we can obtain the remarkable relationship as follows:

$$\Psi(\varsigma) = \frac{1}{(L - M) \sinh(\varsigma) + (L + M) \cosh(\varsigma)}, \tag{6.3}$$

with real potentials $L$, and $M$. For finding $K$ in Eq (6.1), balance between $U^3$ and $U''$ yields

$K = 1$. As a result Eq (6.1) will be converted as

$$U(\varsigma) = A_0 + A_1 \Psi(\varsigma). \tag{6.4}$$

Now, making use of Eqs (6.3), (6.4), and (3.6) and some simple calculation reads

$$\begin{cases} a = -\dfrac{-bk^2A_1^2a_2^2 + 2bkA_1^2a_2 - bA_1^2a_2^2 + 8LMa_1 - bA_1^2}{8LMa_2}, w = \dfrac{bk^2A_1^2a_2 + 8LMka_1 - bkA_1^2 + bA_1^2a_2}{8LMa_2}, b = b, \\[2mm] k = k, L = L, M = M, A_0 = 0, A_1 = A_1, a_1 = a_1, a_2 = a_2, g = \dfrac{2ka_1 - a_2w + a}{a_2k - 1}. \end{cases} \tag{6.5}$$

According to Eqs (6.4) and (6.5), the upcoming result of the mentioned model is obtained:

$$Q_{41}(x, y, t) = \frac{A_1}{(L - M) \sinh(\varsigma) + (L + M) \cosh(\varsigma)} \times \exp(i\delta),$$

where $\delta = -kx + wt + p$ with $w = \frac{ck^2A_1^2l_2 + 8LMkl_1 - ckA_1^2 + cA_1^2l_2}{8LMl_2}$.

## 7 Figure analysis with discussion

Solutions $Q_{26}$ and $Q_{27}$ produce $W$-shaped periodic waves. We depict only $Q_{26}$ (see Figs 1 and 2) for $a_2 = 3$, $k = 2$, $a_1 = M = b = q_0 = q_1 = p = a = 1$. Solutions $Q_3$, $Q_{12}$, $Q_{19}$, $Q_{20}$, $Q_{21}$, $Q_{22}$, $Q_{23}$, $Q_{24}$, and $Q_{25}$ display a single periodic wave with properties that can either amplify or reduce the wave's amplitude. We depict only $Q_{20}$ (see Figs 3–6) for $a_1 = m_1 = \frac{1}{2}$, $a_2 = 2$, $k = 10$, $\vartheta = -\frac{1}{2}$, $G = \rho = p = 1$. We can see that (see Figs 3 and 4) if $b = 1$, the amplitudes of waves rise over time, whereas if $b = -1$, the opposite phenomenon occurs (see Figs 5 and 6). Solutions $Q_{10}$, $Q_{11}$, and $Q_{18}$ display periodic waves with rapidly increasing or decreasing amplitudes. We depict only $Q_{10}$ (see Figs 7–9) for $a_1 = a_2 = 4$, $k = 5$, $\vartheta = -1$, $b = m_1 = G = H = \rho = p = 1$. We can see that (see Fig 7) if $x = -5$, the amplitudes of waves quickly increase over time, whereas if $x = 3$, the opposite phenomenon occurs (see Fig 9). Furthermore, for $x = -1$, the wave's amplitudes rapidly increase and decrease over time (see Fig 8). Solutions $Q_4$ and $Q_{36}$ correspond to double periodic waves, which are depicted by $Q_4$ (see Figs 10–13) for $a_1 = l_1 = G = \rho = p = 1$, $a_2 = \frac{1}{2}$, $k = 20$, $\vartheta = -\frac{1}{2}$, $b = 1$. The real components of $Q_4$ are depicted in Figs 10 and 11, which oscillate frequently with quickly falling amplitudes, then drop to 0 at the origin, and then move again with quickly rising amplitudes. The graphs in Figs

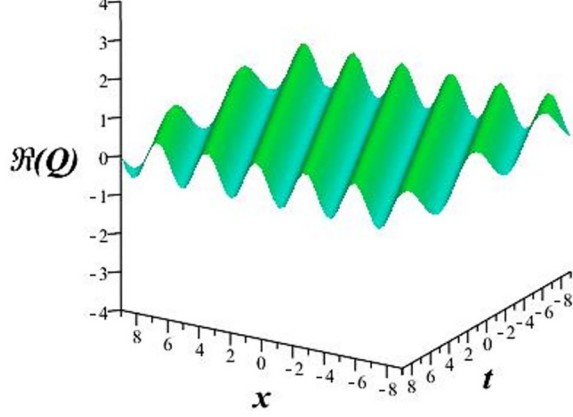

**Fig 1. 3D plot of $Re(Q_{26})$.**

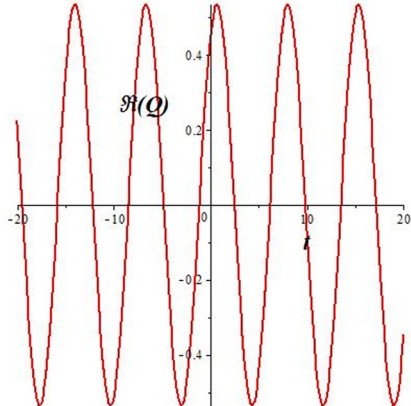

**Fig 2. 2D plot of $Re(Q_{26})$ for $x = 0$.**

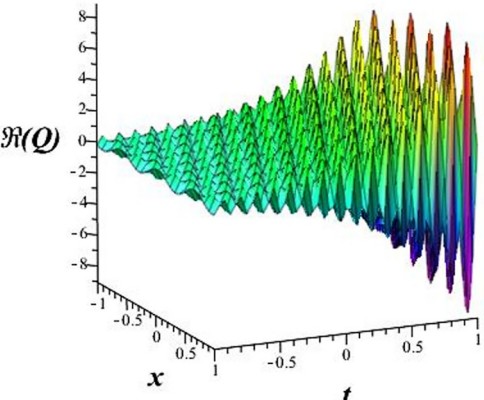

**Fig 3. 3D plot of wave solution $Re(Q_{20})$ for $b = 1$.**

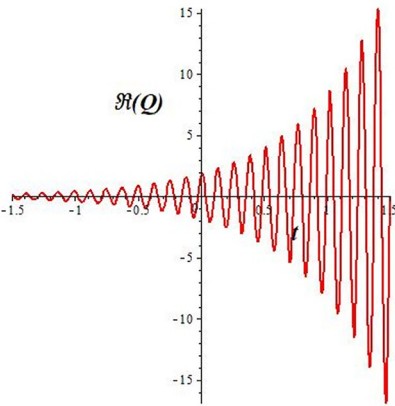

**Fig 4. 2D plot of wave solution $Re(Q_{20})$ for $b = 1$ at $x = 0$.**

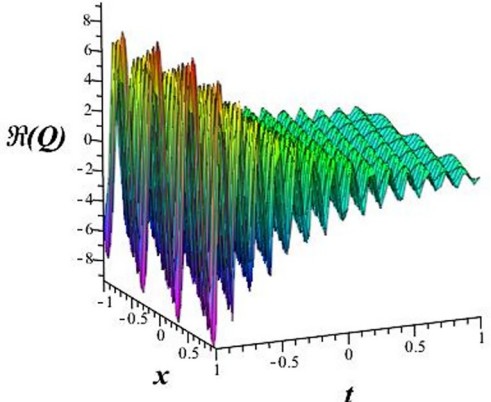

**Fig 5. 3D plot of wave solution $Re(Q_{20})$ for $b = -1$.**

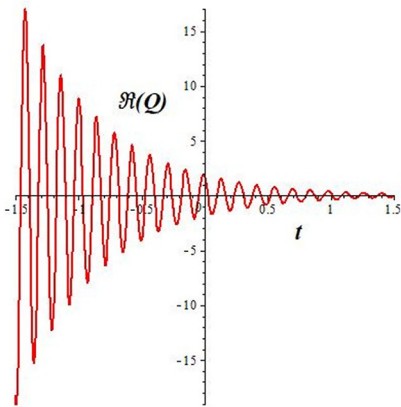

**Fig 6. 2D plot of wave solution $Re(Q_{20})$ for $b = -1$ at $x = 0$.**

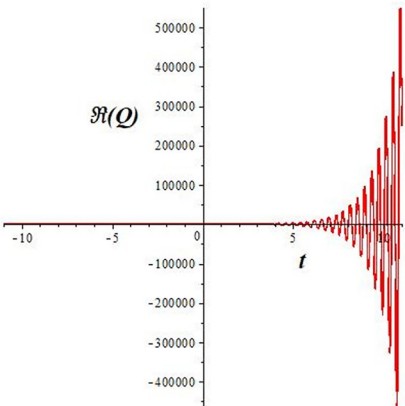

**Fig 7. 2D plot of wave solution $Re(Q_{10})$ for $x = -5$.**

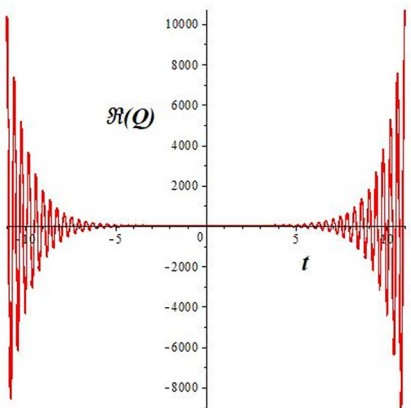

**Fig 8. 2D plot of wave solution $Re(Q_{10})$ for $x = -1$.**

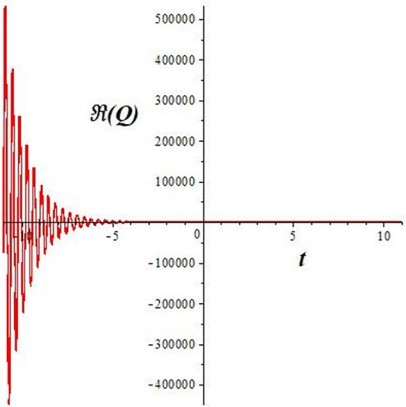

**Fig 9. 2D plot of wave solution $Re(Q_{10})$ for $x = 3$.**

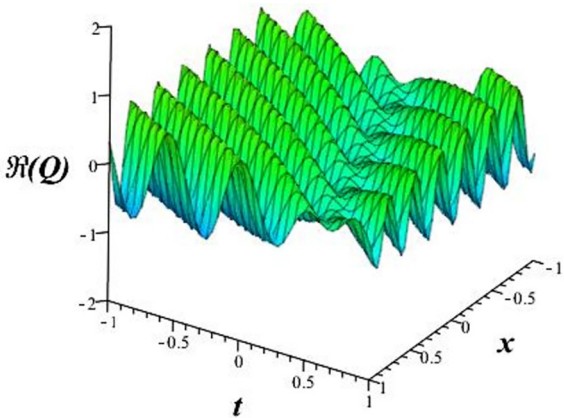

**Fig 10. 3D plot of wave solution $Re(Q_4)$.**

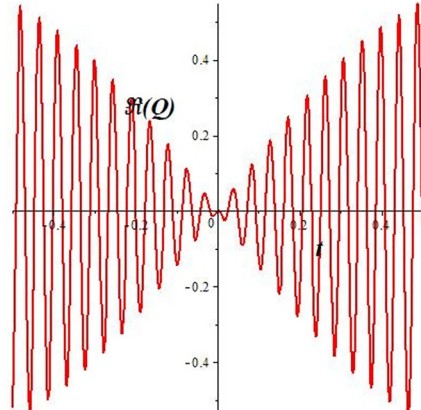

**Fig 11. 2D plot of wave solution $Re(Q_4)$ for $x = -1$.**

12 and 13 are the absolute value solutions of $Q_4$, which correspond to the optical dark soliton solution. Solutions $Q_{41}$ exhibits breather waves (see Figs 14 and 15) and a bright soliton (see Figs 16 and 17) for $a_1 = a_2 = M = b = p = 1$, $L = k = 2$, and $A_1 = 3$. Solutions $Q_7$, and $Q_8$ exhibit multiple breather waves (see Figs 18 and 19) and multiple solitons (see Figs 20 and 21) with symmetric amplitudes, which are plotted by $Q_7$ for $a_1 = a_2 = \rho = 4$, $k = 8$, $\vartheta = b = l_1 = p = 1$, $G = 1.5$. Solutions $Q_5$ and $Q_6$ have multiple breather waves (see Figs 22 and 23) and multiple solitons (see Figs 24 and 25) with asymmetrical amplitudes, which are shown by $Q_5$ for $a_1 = 7$, $a_2 = 5$, $k = 10$, $\vartheta = .7$, $b = l_1 = \rho = p = 1$, $G = 3.5$, $H = 4$. Solutions $Q_{31}$, $Q_{32}$, and $Q_{37}$ have a singular breather wave with constant amplitude, which is plotted by $Q_{37}$ (see Figs 26 and 27) for $a_1 = a_2 = 3$, $k = 2$, $M = b = q_0 = q_1 = p = a = 1$. Solutions $Q_1$, $Q_2$, $Q_9$, $Q_{13}$, $Q_{30}$, $Q_{35}$, and $Q_{40}$ show a singular breather wave with both growing and decreasing amplitudes, as shown by $Q_9$ (see Figs 28 and 29) for $a_1 = 3$, $a_2 = 3$, $k = 5$, $b = l_1 = \rho = 1$, $\vartheta = 0$. Solutions $Q_{14}$, $Q_{15}$, $Q_{16}$, $Q_{17}$, $Q_{28}$, $Q_{29}$, $Q_{33}$, $Q_{34}$, $Q_{38}$, and $Q_{39}$ exhibit singular solutions with multiple breather waves, which are depicted by $Q_{28}$ (see Figs 30–33) for $a_2 = 2$, $k = 5$, $M = -1$, $a_1 = b = p = q_0 = q_1 = a = 1$.

By employing the unified technique, the improved Kudryashov scheme, and the novel Kudryashov approach, the present analysis of the Fokas-Lenells model recovers various

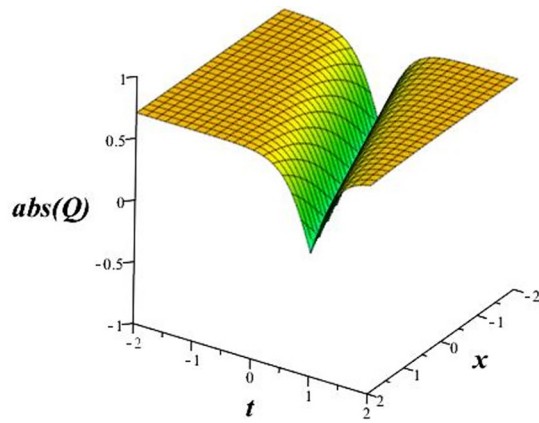

**Fig 12. 3D plot of wave solution $abs(Q_4)$.**

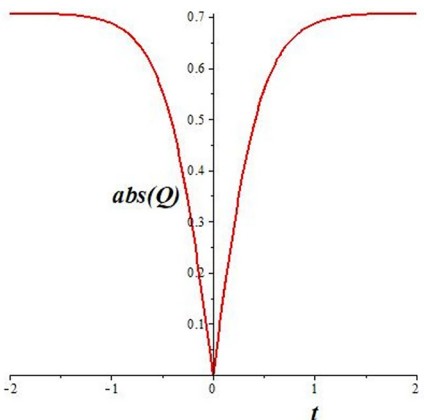

**Fig 13. 2D plot of wave solution $abs(Q_4)$ for $x = -1$.**

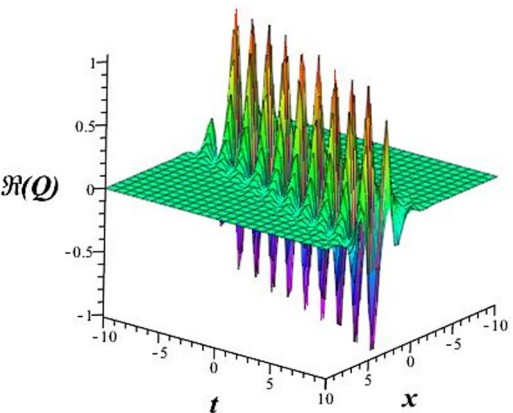

**Fig 14. 3D plot of wave solution $Re(Q_{41})$.**

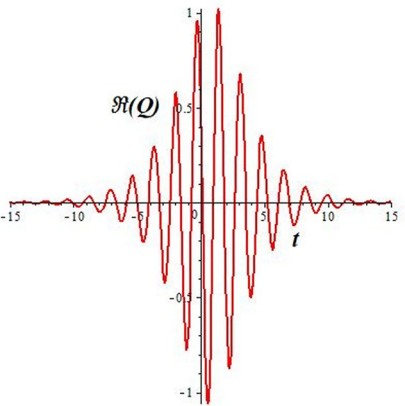

**Fig 15. 2D plot of wave solution $Re(Q_{41})$ for $x = 0$.**

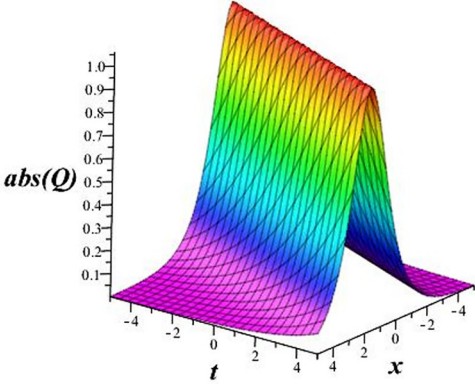

**Fig 16. 3D plot of wave solution** $abs(Q_{41})$.

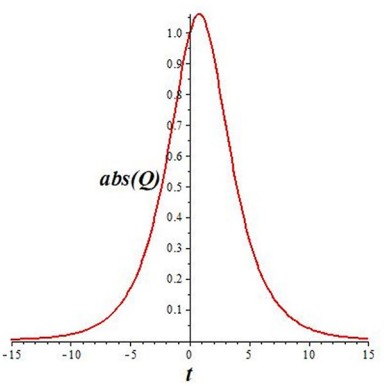

**Fig 17. 2D plot of wave solution** $abs(Q_{41})$ **for** $x = 0$.

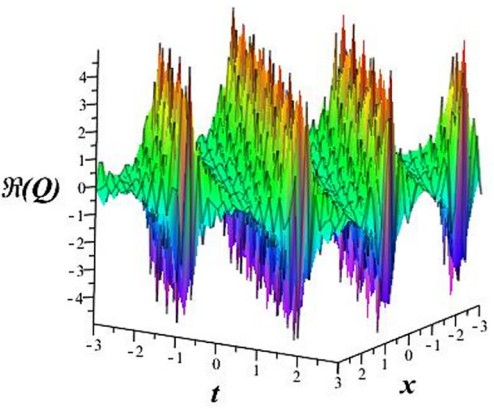

**Fig 18. 3D plot of wave solution** $Re(Q_7)$.

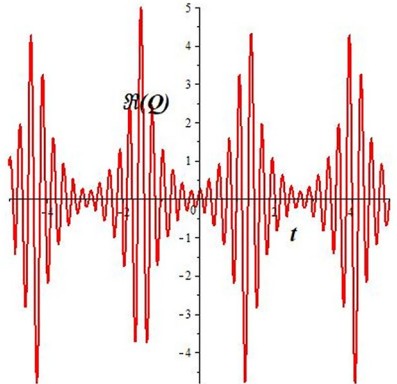

**Fig 19. 2D plot of wave solution $Re(Q_7)$ for $x = -1$.**

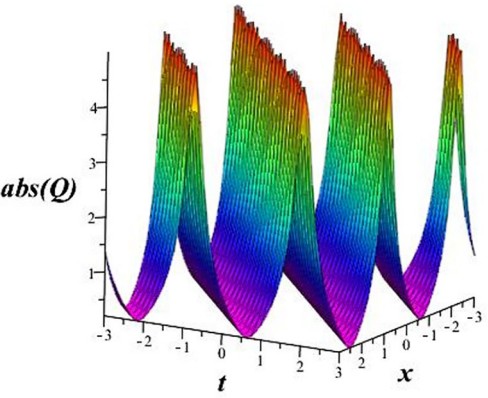

**Fig 20. 3D plot of wave solution $abs(Q_7)$.**

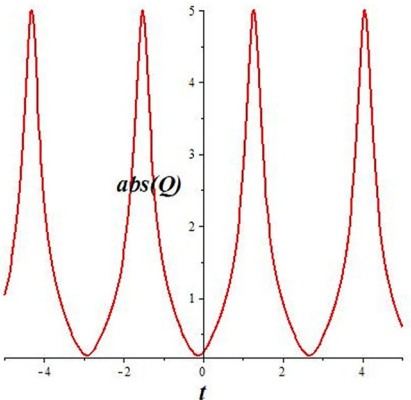

**Fig 21. 2D plot of wave solution $abs(Q_7)$ for $x = -1$.**

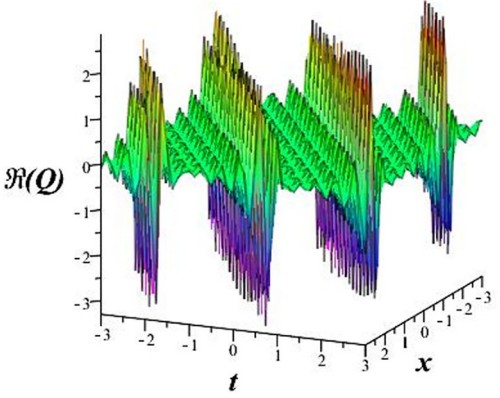

**Fig 22. 3D plot of wave solution** $Re(Q_5)$**.**

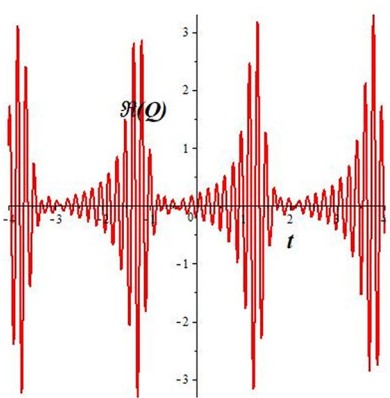

**Fig 23. 2D plot of wave solution** $Re(Q_5)$ **for** $x = 0$**.**

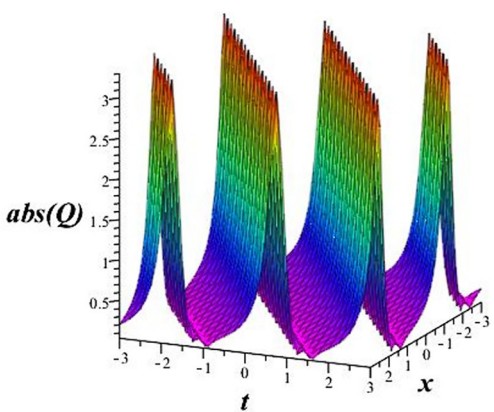

**Fig 24. 3D plot of wave solution** $abs(Q_5)$**.**

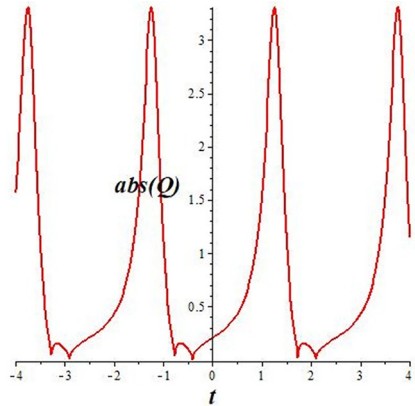

**Fig 25. 2D plot of wave solution $abs(Q_5)$ for $x = 0$.**

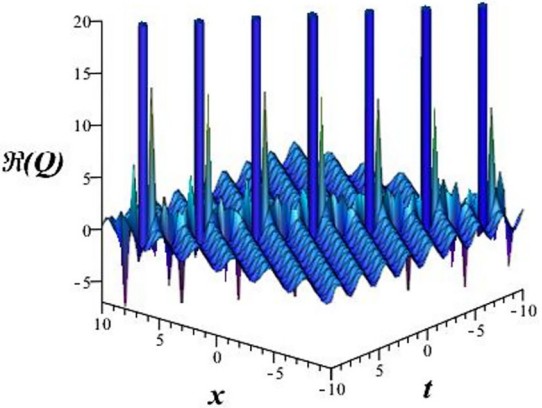

**Fig 26. 3D plot of wave solution $Re(Q_{37})$.**

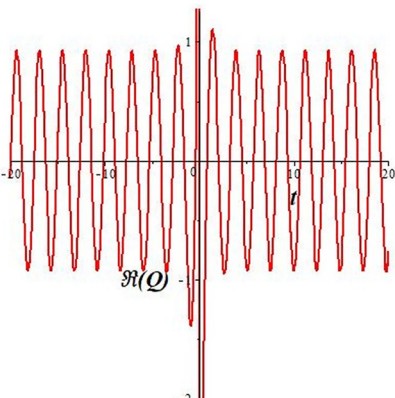

**Fig 27. 2D plot of wave solution $Re(Q_{37})$ for $x = 0$.**

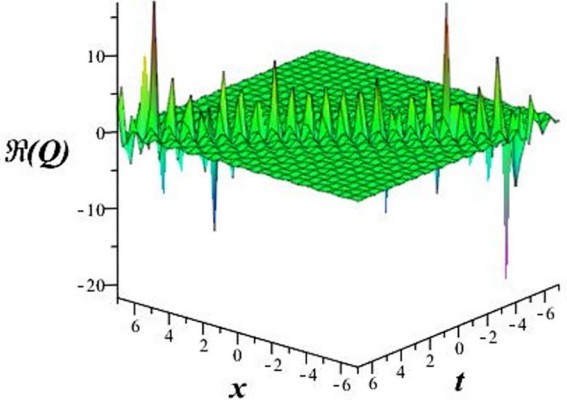

**Fig 28. 3D plot of wave solution $Re(Q_9)$.**

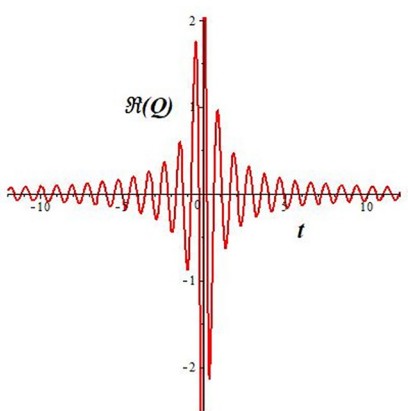

**Fig 29. 2D plot of wave solution $Re(Q_9)$ for $x = -1$.**

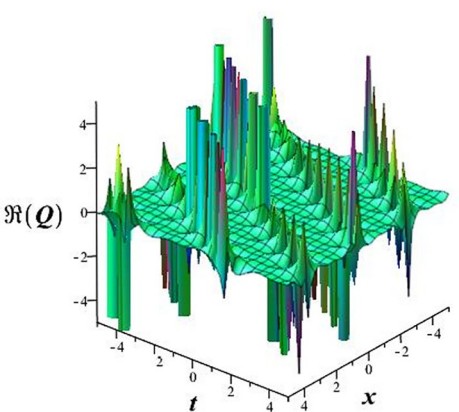

**Fig 30. 3D plot of wave solution $Re(Q_{33})$.**

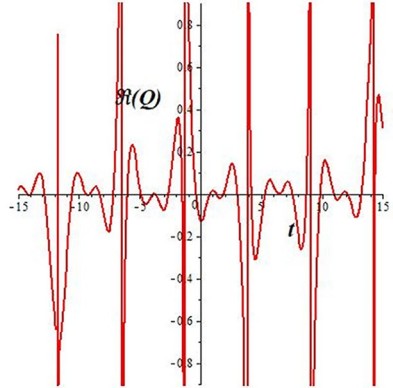

**Fig 31. 2D plot of wave solution** $Re(Q_{33})$ **for** $x = 0$.

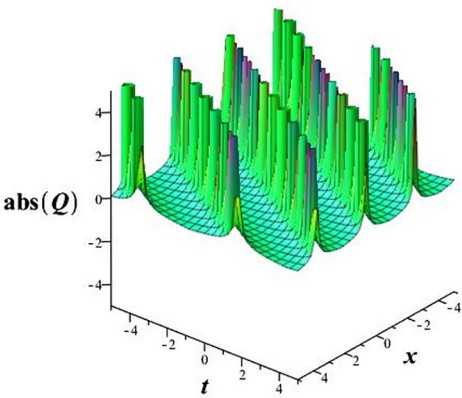

**Fig 32. 3D plot of wave solution** $abs(Q_{33})$.

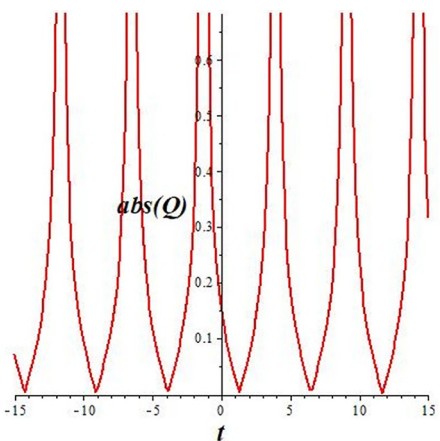

**Fig 33. 2D plot of wave solution** $abs(Q_{33})$ **for** $x = 0$.

novel waveforms and obtains some earlier results when we compare our results to references therein [40, 41]. Our suggested methods are versatile enough to yield solutions to various nonlinear wave equations in hyperbolic, trigonometric, periodic, and exponential forms. Importantly, these solutions can be ready for use without being reduced to any other form. Consequently, there is no necessity to further transform the solutions into other representations. Various dynamic characteristics of outcomes are displayed in 3D, 2D, and density diagrams by setting the parameters involved. The correctness of the calculations was confirmed by reintegrating them into the governing model after wave profiles were created using Maple 18.

## 8 Conclusion

The unified, the improved Kudryashov, and the novel Kudryashov schemes are successfully used in this manuscript to find new waveforms for the Fokas-Lenells dynamical form. The resulting waveforms include periodic W-shaped waves (refer to Figs 1 and 2), periodic waves with gradually increasing amplitudes (observe Figs 3–6), rapidly increasing amplitudes (view Figs 7–9), and double-periodic waves (see Figs 10–13). Single-breather waves (see Figs 14–17) and multi-breather waves with symmetric (see Figs 18–21) and asymmetric (see Figs 22–25) amplitudes are also obtained. Additionally, we obtain singular solutions with single (see Figs 26–29) and multibreather waves (see Figs 30–33). These findings demonstrate that our employed methods are more useful and reliable tools to retrieve optical soliton outcomes for complicated nonlinear models.

## Author Contributions

**Conceptualization:** Harun-Or Roshid.

**Formal analysis:** Harun-Or Roshid.

**Methodology:** Mohammad Safi Ullah.

**Software:** Mohammad Safi Ullah.

**Supervision:** Harun-Or Roshid, M. Zulfikar Ali.

**Validation:** M. Zulfikar Ali.

**Visualization:** M. Zulfikar Ali.

**Writing – original draft:** Mohammad Safi Ullah.

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
