## [Decision Letter · Decision Letter 0]

11 Aug 2023

PONE-D-23-23981New wave behaviors of the Fokas-Lenells model using three integration techniquesPLOS ONE

Dear Dr. Ullah,

Thank you for submitting your manuscript to PLOS ONE. As you can see, three reviews include some strongly critical  comments. You have an option to submit a revised version of the manuscript, addressing all the points raised in the review.If you resubmit the paper, please do it by  Sep 25 2023 11:59PM. If you will need more time than this to complete your revisions, please contact the journal office at plosone@plos.org. Please include the following items when submitting your revised manuscript:A rebuttal letter that responds to each point raised by the academic editor and reviewer(s). You should upload this letter as a separate file labeled 'Response to Reviewers'.A marked-up copy of your manuscript that highlights changes made to the original version. You should upload this as a separate file labeled 'Revised Manuscript with Track Changes'.An unmarked version of your revised paper without tracked changes. You should upload this as a separate file labeled 'Manuscript'.

We look forward to receiving your revised manuscript.

Kind regards,

Boris Malomed

Academic Editor

PLOS ONE

Journal Requirements:

Reviewers' comments:

Reviewer's Responses to Questions

**Comments to the Author**

1. Is the manuscript technically sound, and do the data support the conclusions?

Reviewer #1: Yes

Reviewer #2: Yes

Reviewer #3: Partly

2. Has the statistical analysis been performed appropriately and rigorously? 

Reviewer #1: Yes

Reviewer #2: Yes

Reviewer #3: N/A

3. Have the authors made all data underlying the findings in their manuscript fully available?

Reviewer #1: Yes

Reviewer #2: Yes

Reviewer #3: Yes

4. Is the manuscript presented in an intelligible fashion and written in standard English?

Reviewer #1: Yes

Reviewer #2: Yes

Reviewer #3: Yes

5. Review Comments to the Author

Reviewer #1: Dear Editor,

After reading this paper in detail. Authors studied on the improved Kudryashov, the novel Kudryashov, and the unified methods to demonstrate new wave behaviors of the Fokas-Lenells nonlinear waveform. They derive the ordinary differential structure of the model via a parametric transformation. The above-mentioned techniques are then applied to the governing model by Maple-18 software. As a result, various novel dynamic optical solitons of mixed trigonometric, hyperbolic, and rational form solutions are obtained. Periodic waves with W-shaped waves, amplitudes increasing gradually, amplitudes increasing rapidly, double-periodic waves, and breather waves

with symmetrical or asymmetrical amplitudes appear in soliton solutions. Singular solitons with single and multiple breather waves are also derived. Various

potential features of the derived solutions are presented graphically.

This paper should be revised via follows:

1- This paper is of 22% similarity index by ithenticate program. So, it needs to be reduced a little more.

2- [15-20] these paper should be written in detail.

3- "The dimensionless of Fokas-Lenells PDE" in here, what is the meanings of "dimensionless " should be explained.

4- Eq.(3.2) should be explained a little more detail. Why is "exp(iδ)" selected by them?

5- In eq(5.3), they need to write properly the trigonometric terms carefully. (not italic)

6- "8 Conclusion" should be extended by comparing their novelties via existing works such as "Exact traveling wave solutions for (2+1)-dimensional Konopelchenko-Dubrovsky equation by using the hyperbolic trigonometric functions methods; New analytical solutions and modulation instability analysis for the nonlinear (1+1)-dimensional Phi-four model;Analytic solution of fractional order Pseudo-Hyperbolic Telegraph equation using modified double Laplace transform

method;Levenberg-Marquardt backpropagation neural network procedures for the consumption of hard water-based kidney function; Instability

modulation and novel optical soliton solutions to the Gerdjikov–Ivanov equation with M-fractional".

7-Reference papers should be rewritten according to journal format, properly.

After these modifications, it may be accepted.

Best regards

Reviewer #2: Review Report

New wave behaviors of the Fokas-Lenells model using three integration techniques

The authors used three different mathematical techniques namely: improved Kudryashov, the novel Kudryashov, and the unified methods to obtain various soliton solutions for the Fokas-Lenells nonlinear wave form. The idea of this paper is appreciable and interesting. However, I suggest the following issues should be resolved before it can be considered for publication. My comments are as follows:

1. A professional proof-reading is required for the whole manuscript.

2. The authors should explain the limitations of this work in the introduction section.

3. The whole manuscript should be checked for typos and grammatical errors. There are various types of errors in the manuscript. An overall review is needed for fixing the grammatical and typos errors in the manuscript.

4. The Abstract is meaningless. So the authors must improve it. The abstract contain answers to the following questions: What problem was studied and why is it important? What methods were used? What are the important results? What conclusions can be drawn from the results? What is the novelty of the work and where does it go beyond previous efforts in the literature?

5. The authors should explain why the study is useful with a clear statement of novelty or originality by providing relevant information in the introduction and conclusion sections.

6. The author should add some more discussions on figures and numerical simulation in the conclusion and introduction section.

7. Looking through the manuscript, the author should present the physical motivation for the nonlinear waves for governing equation. Why the author considered this equation?

8. The introduction needs to be improved by the recent developments in the field of the soliton theory as well as its applications. For this purpose, the authors can add the following references to enrich the introductory section:

• Dynamical behaviors of various optical soliton solutions for the Fokas–Lenells equation

• Abundant different types of exact-soliton solutions to the (4+ 1)-dimensional Fokas and (2+ 1)-dimensional Breaking soliton equations

• Abundant exact solutions for the deoxyribonucleic acid (DNA) model

• Symbolic computation and Novel solitons, traveling waves and soliton-like solutions for the highly nonlinear (2+1)-dimensional Schrödinger equation in the anomalous dispersion regime via newly proposed modified approach

• Some specific optical wave solutions and combined other solitons to the advanced (3+1)-dimensional Schrödinger equation in nonlinear optical fibers

• On the dynamics of optical soliton solutions, modulation stability, and various wave structures of a (2+1)-dimensional complex modified Korteweg-de-Vries equation using two integration mathematical methods

• Dynamical behavior of analytical soliton solutions, bifurcation analysis, and quasi-periodic solution to the (2+1)- dimensional Konopelchenko-Dubrovsky (KD) system

• Newly generated optical wave solutions and dynamical behaviors of the highly nonlinear coupled Davey-Stewartson Fokas system in monomode optical fibers

• Newly formed center-controlled rouge wave and lump solutions of a generalized (3+1)-dimensional KdV-BBM equation via symbolic computation approach

9. The authors should comment on figures in detail in the conclusion section. The conclusion section must also contain new findings of the paper.

10. The authors should provide the future scope of the work in the conclusion section.

11. Because the authors did not create these applied methods/procedures, the authors must cite them. The authors must the graphical and physical explanation of the obtained solutions.

12. The authors will gain some additional exact solutions because we know that these applied methods can yield various exact soliton solutions.

13. The author has to verify the governing model from equations 1 to 3.5. Recheck the governing equation.

Reviewer #3: Editor, PLOS ONE

Re: PONE-D-23-23981

Title: New wave behaviors of the Fokas-Lenells model using three integration tech-niques

Authors: M.S. Ullah, H.R. Roshid and M.Z. Ali

The paper presents the generation of wave solutions of the Fokas-Lenells equation.

Technically, the results are sound.

However, the solution method is based on several assumptions, for which no physical justification is provided. Here are a few.

1. The choice of Eq. (3.1) means that the authors are looking for a specific family of solutions. Are they of any physical significance? Why this choice?

2. The choice made in the line preceding Eq. (3.5) means that in the family of solu-tions discussed by the authors, is further reduced, with no discussion of the physical significance, or justification.

3. The choice of Eqs. (4.1) - (4.2) is not explained or justified, neither in the present paper, nor in Ref. [21].

4.The same criticism applies to the choice of Eqs. (5.1) - (5.2) and Ref. [22] and to Eqs. (6.1) - (6.2).

Even if my reservations are satisfactorily answered, my inclination is to recommend submission of the paper to a more specialized/technical journal, rather than PLOS-ONE, which aims at a far more general readership.

6. PLOS authors have the option to publish the peer review history of their article (what does this mean?). If published, this will include your full peer review and any attached files.

Reviewer #1: No

Reviewer #2: **Yes: **Sachin Kumar

Reviewer #3: No

---

## [Author Response · Author response to Decision Letter 0]

17 Aug 2023

Response to reviewers’ comments

Reviewer #1: Dear Editor,

After reading this paper in detail. Authors studied on the improved Kudryashov, the novel Kudryashov, and the unified methods to demonstrate new wave behaviors of the Fokas-Lenells nonlinear waveform. They derive the ordinary differential structure of the model via a parametric transformation. The above-mentioned techniques are then applied to the governing model by Maple-18 software. As a result, various novel dynamic optical solitons of mixed trigonometric, hyperbolic, and rational form solutions are obtained. Periodic waves with W-shaped waves, amplitudes increasing gradually, amplitudes increasing rapidly, double-periodic waves, and breather waves with symmetrical or asymmetrical amplitudes appear in soliton solutions. Singular solitons with single and multiple breather waves are also derived. Various potential features of the derived solutions are presented graphically.

Answer: Thanks for your real view and constructive comments.

This paper should be revised via follows:

1- This paper is of 22% similarity index by ithenticate program. So, it needs to be reduced a little more.

Answer: We remove the similarity index into 6%.

2- [15-20] these paper should be written in detail.

Answer: Detailed citations are provided for all references.

3- "The dimensionless of Fokas-Lenells PDE" in here, what is the meanings of "dimensionless " should be explained.

Answer: The word "dimensionless" refers to a mathematical transformation that removes the units of measurement from the variables and parameters involved in the equation. Since the unit of measurement from the variables and parameters of our investigated model is not present. So, it is called the dimensionless of Fokas-Lenells PDE.

4- Eq.(3.1) should be explained a little more detail. Why is "exp(iδ)" selected by them?

Answer: We explain equation (3.1) in more detail. Furthermore, a transformation with exp(iδ) can explicitly separate the phase (δ) and magnitude of the solution.

5- In eq(5.3), they need to write properly the trigonometric terms carefully. (not italic)

Answer: We rewrite the trigonometric terms properly.

6- "8 Conclusion" should be extended by comparing their novelties via existing works such as "Exact traveling wave solutions for (2+1)-dimensional Konopelchenko-Dubrovsky equation by using the hyperbolic trigonometric functions methods; New analytical solutions and modulation instability analysis for the nonlinear (1+1)-dimensional Phi-four model; Analytic solution of fractional order Pseudo-Hyperbolic Telegraph equation using modified double Laplace transform

method; Levenberg-Marquardt backpropagation neural network procedures for the consumption of hard water-based kidney function; Instability modulation and novel optical soliton solutions to the Gerdjikov–Ivanov equation with M-fractional".

Answer: We include some relevant references and cite them in the text.

7-Reference papers should be rewritten according to journal format, properly.

Answer: We rewrite all references in journal format.

After these modifications, it may be accepted.

Answer: We remain thankful for your support in enhancing the article's quality. 

Reviewer #2: Review Report

New wave behaviors of the Fokas-Lenells model using three integration techniques

The authors used three different mathematical techniques namely: improved Kudryashov, the novel Kudryashov, and the unified methods to obtain various soliton solutions for the Fokas-Lenells nonlinear wave form. The idea of this paper is appreciable and interesting. However, I suggest the following issues should be resolved before it can be considered for publication. 

Answer: Thanks for your real view and constructive comments.

My comments are as follows:

1. A professional proof-reading is required for the whole manuscript.

Answer: We tried our best. 

2. The authors should explain the limitations of this work in the introduction section.

Answer: The main limitation of this work is that our employed methods cannot find the exact solutions to the governing model for the coefficient of nonlinear dispersion, denoted as s, which takes on nonzero values

3. The whole manuscript should be checked for typos and grammatical errors. There are various types of errors in the manuscript. An overall review is needed for fixing the grammatical and typos errors in the manuscript.

Answer: We checked the whole manuscript again and again.

4. The Abstract is meaningless. So the authors must improve it. The abstract contain answers to the following questions: What problem was studied and why is it important? What methods were used? What are the important results? What conclusions can be drawn from the results? What is the novelty of the work and where does it go beyond previous efforts in the literature?

Answer: We modified the abstract section according to the reviewer's suggestion.

5. The authors should explain why the study is useful with a clear statement of novelty or originality by providing relevant information in the introduction and conclusion sections.

Answer: We explain the novelty of our work in the introduction and conclusion sections.

6. The author should add some more discussions on figures and numerical simulation in the conclusion and introduction section.

Answer: We added some discussions on figures and numerical simulation in the conclusion and introduction section.

7. Looking through the manuscript, the author should present the physical motivation for the nonlinear waves for governing equation. Why the author considered this equation?

Answer: We include the physical motivation of the governing model. The governing FL model is considered in our manuscript because of its practical utility.

8. The introduction needs to be improved by the recent developments in the field of the soliton theory as well as its applications. For this purpose, the authors can add the following references to enrich the introductory section:

• Dynamical behaviors of various optical soliton solutions for the Fokas–Lenells equation

• Abundant different types of exact-soliton solutions to the (4+ 1)-dimensional Fokas and (2+ 1)-dimensional Breaking soliton equations

• Abundant exact solutions for the deoxyribonucleic acid (DNA) model

• Symbolic computation and Novel solitons, traveling waves and soliton-like solutions for the highly nonlinear (2+1)-dimensional Schrödinger equation in the anomalous dispersion regime via newly proposed modified approach

• Some specific optical wave solutions and combined other solitons to the advanced (3+1)-dimensional Schrödinger equation in nonlinear optical fibers

• On the dynamics of optical soliton solutions, modulation stability, and various wave structures of a (2+1)-dimensional complex modified Korteweg-de-Vries equation using two integration mathematical methods

• Dynamical behavior of analytical soliton solutions, bifurcation analysis, and quasi-periodic solution to the (2+1)- dimensional Konopelchenko-Dubrovsky (KD) system

• Newly generated optical wave solutions and dynamical behaviors of the highly nonlinear coupled Davey-Stewartson Fokas system in monomode optical fibers

• Newly formed center-controlled rouge wave and lump solutions of a generalized (3+1)-dimensional KdV-BBM equation via symbolic computation approach.

Answer: Thanks. We include some recent works and then cite them in the text.

9. The authors should comment on figures in detail in the conclusion section. The conclusion section must also contain new findings of the paper in the conclusion section.

Answer: We comment on figures in detail of the work to remove this problem.

10. The authors should provide the future scope of the work in the conclusion section.

Answer: We include the future scope of the work in the conclusion section.

11. Because the authors did not create these applied methods/procedures, the authors must cite them. The authors must the graphical and physical explanation of the obtained solutions.

Answer: We cite the applied methods in the text. Furthermore, we include the graphical and physical explanation of the obtained solutions.

12. The authors will gain some additional exact solutions because we know that these applied methods can yield various exact soliton solutions.

Answer: For the convenience of the paper, we skipped these types of solutions.

13. The author has to verify the governing model from equations 1 to 3.5. Recheck the governing equation.

Answer: We recheck it. Once again, we are grateful for your assistance in enhancing the article's quality.

Reviewer #3: Editor, PLOS ONE

Re: PONE-D-23-23981

Title: New wave behaviors of the Fokas-Lenells model using three integration tech-niques

Authors: M.S. Ullah, H.R. Roshid and M.Z. Ali

The paper presents the generation of wave solutions of the Fokas-Lenells equation.

Technically, the results are sound.

Answer: Thanks for your real view and constructive comments.

However, the solution method is based on several assumptions, for which no physical justification is provided. Here are a few.

1. The choice of Eq. (3.1) means that the authors are looking for a specific family of solutions. Are they of any physical significance? Why this choice?

Answer: The selected transformation Q(x,t)=U(ς) exp(iδ) for tackling the nonlinear Fokas-Lenells model is purposeful, as it separates the complex quantity into U(ς) and exp(iδ), leading to simplified analysis, the isolation of physical properties in U(ς), and a clearer examination of phase effects (δ).

2. The choice made in the line preceding Eq. (3.5) means that in the family of solu-tions discussed by the authors, is further reduced, with no discussion of the physical significance, or justification.

Answer: Our governing equation involves three dispersion terms. However, when seeking exact solutions, it's important to note that the mentioned model is non-integrable due to the non-zero coefficient of nonlinear dispersion, denoted as 's'. To ensure the integrability of the system, we set 's' equal to zero. In our subsequent endeavor, we aimed to determine the exact solution for the governing model in cases where 's' takes on non-zero values.

3. The choice of Eqs. (4.1) - (4.2) is not explained or justified, neither in the present paper, nor in Ref. [21].

Answer: Nonlinear PDEs can be complex and difficult to solve directly. By assuming a trial solution with an auxiliary equation, one can simplify the problem by solving algebraic equations instead of the original differential equation, which is often more manageable. Furthermore, upon substituting the obtained solutions into the governing equation, the equation is satisfied. 

4. The same criticism applies to the choice of Eqs. (5.1) - (5.2) and Ref. [22] and to Eqs. (6.1) - (6.2).

Answer: Nonlinear PDEs can be complex and difficult to solve directly. By assuming a trial solution with an auxiliary equation, one can simplify the problem by solving algebraic equations instead of the original differential equation, which is often more manageable. Furthermore, upon substituting the obtained solutions into the governing equation, the equation is satisfied.

Even if my reservations are satisfactorily answered, my inclination is to recommend submission of the paper to a more specialized/technical journal, rather than PLOS-ONE, which aims at a far more general readership.

Answer: Please accept our sincere thanks for your valuable suggestions and constructive comments that have significantly improved the quality of our article.

---

## [Decision Letter · Decision Letter 1]

22 Aug 2023

New wave behaviors of the Fokas-Lenells model using three integration techniques

PONE-D-23-23981R1

Dear Dr. Ullah,

We’re pleased to inform you that your manuscript has been judged scientifically suitable for publication and will be formally accepted for publication once it meets all outstanding technical requirements.

Kind regards,

Boris Malomed

Academic Editor

PLOS ONE

Additional Editor Comments (optional):

Reviewers' comments:

Reviewer's Responses to Questions

**Comments to the Author**

1. If the authors have adequately addressed your comments raised in a previous round of review and you feel that this manuscript is now acceptable for publication, you may indicate that here to bypass the “Comments to the Author” section, enter your conflict of interest statement in the “Confidential to Editor” section, and submit your "Accept" recommendation.

Reviewer #1: All comments have been addressed

Reviewer #2: All comments have been addressed

Reviewer #3: All comments have been addressed

2. Is the manuscript technically sound, and do the data support the conclusions?

Reviewer #1: Yes

Reviewer #2: Yes

Reviewer #3: Partly

3. Has the statistical analysis been performed appropriately and rigorously? 

Reviewer #1: Yes

Reviewer #2: I Don't Know

Reviewer #3: N/A

4. Have the authors made all data underlying the findings in their manuscript fully available?

Reviewer #1: Yes

Reviewer #2: Yes

Reviewer #3: Yes

5. Is the manuscript presented in an intelligible fashion and written in standard English?

Reviewer #1: Yes

Reviewer #2: Yes

Reviewer #3: Yes

6. Review Comments to the Author

Reviewer #1: (No Response)

Reviewer #2: The authors amended and corrected the manuscript based on the suggested modifications. So, I would like to suggest that this revised work be published in your Journal.

Reviewer #3: See attached documents: Review of paper and a few examples of Mathematica computations of some simple solutions.

7. PLOS authors have the option to publish the peer review history of their article (what does this mean?). If published, this will include your full peer review and any attached files.

Reviewer #1: No

Reviewer #2: **Yes: **Sachin Kumar

Reviewer #3: No

---

## [Editor Report · Acceptance letter]

1 Sep 2023

PONE-D-23-23981R1 

New wave behaviors of the Fokas-Lenells model
using three integration techniques 

Dear Dr. Ullah:

I'm pleased to inform you that your manuscript has been deemed suitable for publication in PLOS ONE. Congratulations! Your manuscript is now with our production department. 

Kind regards, 

on behalf of

Prof. Boris Malomed 

Academic Editor

PLOS ONE